# Regulation of transcription elongation anticipates alternative gene expression strategies across the cell cycle

Douglas Maya-Miles[1,2], José García-Martínez[3], Ildefonso Cases[4], Rocío Pasión[5], Jesús de la Cruz[1,5], José Enrique Pérez-Ortín[3]*, María de la Cruz Muñoz-Centeno[1,5]*, Sebastián Chávez[1,5]*

1 Instituto de Biomedicina de Sevilla, Universidad de Sevilla-CSIC-Hospital Universitario Virgen del Rocío, Seville, Spain, 2 Centro de Investigación Biomédica en Red (CIBEREHD), Sevilla, Spain, 3 Instituto de Biotecnología y Biomedicina (BIOTECMED), Facultad de Biológicas, Universitat de València, Burjassot, Spain, 4 Unidad de Bioinformática, Centro Andaluz de Biología del Desarrollo, CSIC-Universidad Pablo de Olavide-Junta de Andalucía, Seville, Spain, 5 Departamento de Genética, Facultad de Biología, Universidad de Sevilla, Seville, Spain

* jose.e.perez@uv.es (JEPO), mcmunoz@us.es (MC), schavez@us.es (SC)

## Abstract

A growing body of evidence supports the idea that RNA polymerase II (RNAP II) activity during transcription elongation can be regulated to control transcription rates. Using genomic run-on and RNAP II chromatin immunoprecipitation, we measured both active and total RNAP II across the bodies of genes at three different stages of the mitotic cell cycle in *Saccharomyces cerevisiae*: G1, S, and G2/M. Comparison of active and total RNAP II levels at these stages revealed distinct patterns of transcription elongation control throughout the cell cycle. Previously characterized cycling genes were associated with some of these elongation patterns. A cluster of genes with highly divergent genomic run-on and RNAP II chromatin immunoprecipitation patterns was notably enriched in genes related to ribosome biogenesis and the structural components of the ribosome. We confirmed that the expression of ribosome biogenesis mRNAs increases after G1 but decreases following mitosis. Finally, we analyzed the contribution of mRNA stability to each cluster and found that a coordinated regulation of RNAP II activity and mRNA decay is necessary to fully understand the alternative strategies of gene expression across the cell cycle.

## Introduction

Phenotypic plasticity in living organisms is driven by the regulation of genome expression. This regulation occurs at all levels of the gene expression pathway, from gene transcription to protein degradation (reviewed in [1]). Transcription is the most highly regulated step in most organisms and under most conditions [2–4], although other steps also play significant roles during physiological dynamic changes [5].

**Data availability statement:** All relevant data are within the manuscript and its Supporting Information files. Raw data produced in the experiments described in this work are publicly available in the Gene Expression Omnibus repository (GSE277924): https://www.ncbi.nlm.nih.gov/geo/query/acc.cgi?acc=GSE277924.

**Funding:** "This work was supported by grants PID2020-112853GB-C31 to JEP-O, PID2022-136564NB-I00 to JdlC, and PID2023-148037NB-C21 to SC and MCM-C, funded by MICIU/AEI/10.13039/501100011033/ERDF, EU, and CIAICO/2022/237 funded by Generalitat Valenciana to JEP-O. The funders had no role in study design, data collection and analysis, decision to publish, or preparation of the manuscript".

**Competing interests:** The authors have declared that no competing interests exist.

Protein-coding gene transcription is carried out by RNA polymerase II (RNAP II), which initiates at gene promoters, elongates pre-mRNAs along gene bodies, and terminates once pre-mRNAs are cleaved and polyadenylated. Historically, most of our understanding of transcriptional regulation has focused on the initiation phase. However, in recent decades, it has become clear that post-initiation regulation is a crucial component of genome expression control [6,7].

In higher eukaryotes, transcriptional elongation is primarily regulated through promoter-proximal pausing. In the budding yeast *Saccharomyces cerevisiae*, this mechanism does not occur. However, other forms of elongation regulation are present across the yeast genome [8], which contains all the necessary factors, including chromatin remodeling complexes, histone chaperones, and histone-modifying enzymes, that govern eukaryotic transcription elongation [9,10].

During elongation, RNAP II can backtrack on its DNA template, stably blocking mRNA elongation [11]. Backtracked RNAP II requires RNA cleavage to resume transcription [12], a process stimulated by TFIIS [11]. Using RNAP II chromatin immunoprecipitation (RNAp ChIP) and genomic run-on (GRO) assays to compare total and catalytically active elongating polymerases, respectively, we previously demonstrated that certain gene regulons, such as those related to ribosomal proteins, ribosome biogenesis, and mitochondria, are regulated at the elongation level during carbon source changes. Notably, ribosomal protein genes are particularly susceptible to backtracking [8].

The *S. cerevisiae* genome undergoes substantial changes in gene expression throughout the cell cycle. Approximately 10% of its mRNAs exhibit cell-cycle periodicity [13]. Some reports suggest this proportion may be as high as 29%, with 37% of the proteome being regulated in a cell-cycle-dependent manner [14], though other studies challenge these findings (reviewed in [15]; see also [16]).

In this study, we investigated whether regulation of RNAP II elongation activity plays a role in genome-wide control of gene expression across the cell cycle. Our results show that this regulation is indeed present in a significant portion of the genome. Genes encoding ribosome biogenesis factors appear to be particularly subject to elongation regulation and tend to increase the proportion of active RNAP II molecules just before the onset of mitosis. Furthermore, we found that distinct groups of genes with regulated RNAP II elongation activity also exhibit concomitant control of mRNA stability throughout the cell cycle, suggesting a potential link between these two components of gene regulation.

## Materials and methods

### Cell cycle synchronization

BY4741 *S. cerevisiae* cells, transformed with plasmid pRS316 to become Ura+, were grown to mid-log phase ($OD_{600}=0.25$) in minimal complete (SC) medium and synchronized at G1 (START) with 1 µg/ml of alpha factor. At this stage, the percentage of unbudded cells in all three experiments was greater than 90%. The cultures were then washed in fresh media to remove the pheromone and released into new

media. Samples for total RNAP II (RNAp ChIP), active RNAP II (GRO), FACS, and microscopy were taken at 0, 30, and 60 minutes.

To determine which cell cycle stage was enriched at each time point, we first performed flow cytometry and budding index analysis. The combination of these two analyses revealed that, at time 0, most cells remained in G1 (unbudded cells with or without shmoo formation and a 1C DNA content), with very few having completed mitosis by 60 minutes (indicated by two differentiated nuclear masses). These two analyses are useful for distinguishing the G1 (no bud or shmoo) and mid-late M phase (where the nucleus has started to split) populations, but they are less effective at differentiating between S, G2, and early M phases. Median levels of total and active RNAP II at each time point, based on well-established cell cycle genes [17,18], were used to more accurately infer which cell cycle stages were enriched at each time point analyzed (Fig 1A, 1C).

## Genomic run-on

Genomic run-on (GRO) was performed as described in [19] and modified in [20]. Briefly, GRO detects genome-wide, actively elongating RNAP II by macroarray hybridization, with the density of RNAP II per gene used as a measure of its synthesis rate. The protocol also allows for the measurement of mRNA levels for all genes. mRNA half-lives are calculated by dividing the mRNA amount by the synthesis rate, assuming steady-state conditions for the transcriptome. The total synthesis rate was determined by summing the individual gene synthesis rates.

For the procedure, $5 \times 10^8$ exponentially growing cells were permeabilized by washing twice with 0.5% N-lauryl sarcosine sodium sulfate (sarkosyl), followed by recovery through low-speed centrifugation. In vivo transcription was then carried out in a solution containing 120 µL of 2.5X transcription buffer (50 mM Tris-HCl, pH 7.7, 500 mM KCl, 80 mM $MgCl_2$), 16 µL of AGC mix (10 mM each of CTP, ATP, and GTP), 6 µL of DTT (0.1 M), 16 µL of [α33P] UTP (3000 Ci/mmol, 10 µCi/µL), and distilled water up to 300 µL. The mixture was incubated for 5 minutes at 30ºC to facilitate transcription elongation and was subsequently halted by adding 1 mL of cold distilled water. Non-incorporated radioactive nucleotides were removed by two centrifugation wash cycles.

Total RNA was extracted using the Fast-Prep (Bio101 Inc.) device. Cells were resuspended in 500 µL of LETS buffer (0.1 M LiCl, 10 mM EDTA, 0.2% SDS, 10 mM Tris-HCl, pH 7.4), along with 200 µL of glass beads and 500 µL of water-saturated acid phenol. Contaminants were removed by chloroform extraction, and labeled RNA was precipitated with 0.1 volume of 5 M LiCl and 2.5 volumes of cold ethanol for a minimum of 2 hours at -20ºC. The labeled RNA, obtained after centrifugation and washing, was quantified spectrophotometrically. All in vivo labeled RNA was used for hybridization ($0.35–3.5 \times 10^7$ dpm).

## Chromatin immunoprecipitation of RNA polymerase II (RNAp ChIP)

RNAp ChIP was performed as described in [21]. $5 \times 10^8$ cells were crosslinked with formaldehyde (FC 1%) for 15 minutes at room temperature. Cross-link was quenched with glycine (FC 125 mM) and washed 4 times with 30 mL ice-cold TBS buffer (20 mM Tris-HCl, 140 mM NaCl, pH 7.5) and frozen. Cells were thawed on ice and resuspended in 300 µL lysis buffer [50 mM HEPES-KOH pH 7.5, 140 mM NaCl, 1 mM EDTA, 1% Triton X-100, 0.1% sodium deoxycholate, 1 mM PMSF (Phenylmethylsulfonyl fluoride), 1 mM benzamidine and 1 pill of protease inhibitor cocktail (Roche Diagnostics, Mannheim, Germany) dissolved in every 50 ml of buffer)]. The equivalent of 0.3 ml of frozen glass beads (425–600 mm; Sigma-Aldrich, St Louis, MO, USA) Cell lysis was carried out in a Fast-Prep (Bio101 Inc.) device by 2 cycles of 30 seconds. Samples were cooled down 5 minutes in ice between cycles. Samples were then sonicated in ice-cold water using a Bioruptor (Diagenode SA, Liège, Belgium) with 15′ pulses at High output (200W) during 2 cycles of 15 minutes. DNA fragments obtained ranged from 200 and 500 bp. Cell debris was removed by centrifugation at 12,000 rpm at 4°C for 5 minutes. An aliquot of 10 µL of this whole cell extract (WCE) was kept as positive control. 50 µL of Dynabeads Protein

**A.**

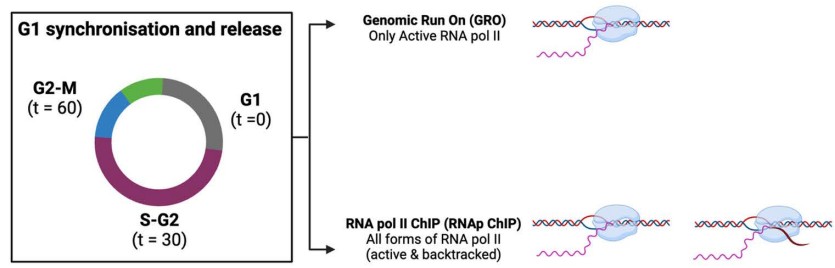

**B.**

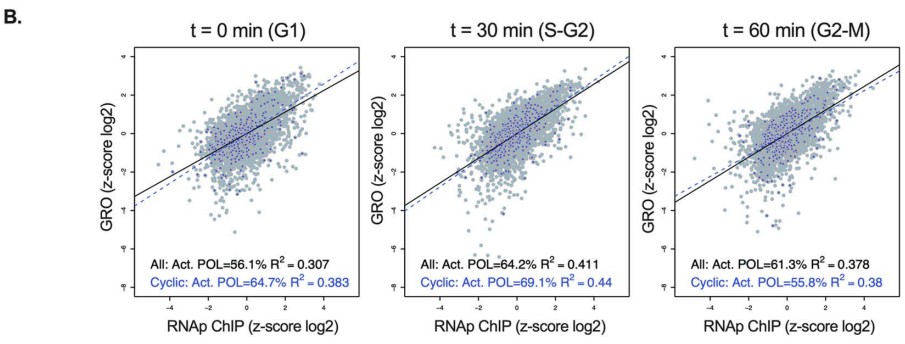

**C.**

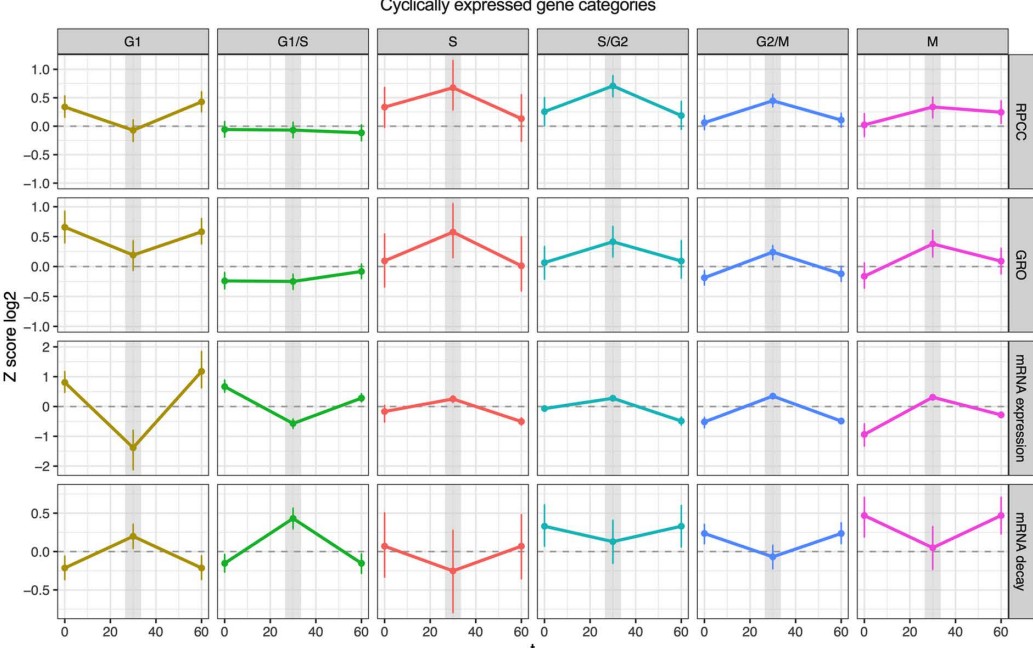

**Fig 1. Distribution of Total and Active RNA Polymerase II at Genes During the Transition from G1 to G2.** A) Schematic representation of the time points and samples taken for the analysis of total and active RNAP II. B) Plot representing total (RNAp ChIP) and active RNAP II (GRO) present at each

gene at each time point analyzed. Black line: Overall tendency. Dashed blue line: Cycling genes. C) Behavior of total (RNAp ChIP) and active RNAP II (GRO) of cyclically expressed genes, classified according to the phase of maximal expression (G1, G1/S, S, S/G2, G2/M, and M) using CycleBase 3.0 as a reference (S1 Table). Total mRNA expression and mRNA decay values from Eser et al. [17] are also shown.

G (Dynal Invitrogen Corp., Carlsbad, CA, USA) were washed with 5 mg/ml bovine serum albumin (BSA) in PBS buffer (140 mM NaCl, 2.7 mM KCl, 10 mM $Na_2HPO_4$, 1.8 mM $KH_2PO_4$, pH 7,4) and incubated with anti-Rpb3 antibodies (ab81859; Abcam). Beads were washed with PBS/BSA to remove unbound antibody and incubated for 1.5–2 hours in agitation to enrich the samples in the DNA fragments specifically cross-linked to the RNAP II. Beads were washed twice with lysis buffer, twice with 360 mM NaCl lysis buffer, twice with wash buffer (10 mM Tris-HCl pH 8.0, 250 mM LiCl, 0.5% Nonidet P-40, 0.5% sodium deoxycholate, 1 mM EDTA pH 8.0, 1 mM PMSF, 1 mM benzamidine and 1 pill of protease inhibitor cocktail/25 mL (Roche), and once with TE (10 mM Tris-HCl, pH 8.0, 1 mM EDTA). Two successive washes were performed with 30 µL of elution buffer (50 mM Tris-HCl pH 8.0, 10 mM EDTA, 1% SDS) by incubating for 10 minutes at 65°C each time. The final volume of all the samples (IP, NA and WCE) was raised to 300 µL with TE and treated overnight at 65°C to reverse the cross-linking. Proteins were degraded by adding 150 µg of proteinase K for 1h 30min at 37°C. DNA was purified using QIAquick PCR purification columns (Qiagen Inc., Valencia, CA, USA). Ligation Mediated PCR [3] was used for DNA amplification. Briefly, DNA was blunted by T4 phage DNA polymerase in a reaction volume of 124 µL (T4 DNA Pol buffer, 40 µg/µL BSA, 80 µM dNTPs, 0.6 U T4 DNA Polymerase (Roche, Basel, Switzerland). The reaction was allowed to proceed for 20 minutes at 12°C. After phenol/chloroform/isoamylic alcohol extraction, DNA was ethanol-precipitated in the presence of 12 µg of glycogen and was ligated in a final volume of 50 µL with the annealed linkers oJW102 and oJW103 (1.5 µM of each primer). The reaction was carried out overnight at 16°C and ligated DNA was precipitated and resuspended in 25 µL milliQ water. The ligated DNA was dissolved in a final volume of 40 µL (1x DNA pol buffer: 2 mM $MgCl_2$, 0.25 mM dNTPs, 1.25 µM oligonucleotide oJW102). The reaction was started by incubating for 2 minutes at 55°C, then pausing to add 10 µL of the reaction mix (1x DNA pol buffer, 2 mM $MgCl_2$ and 5 U DNA polymerase from Biotools, Madrid, Spain). The program was resumed for 5 minutes at 72°C, 2 minutes at 95°C and 33 cycles of 30 s at 95°C, 30 s at 55°C and 2 minutes at 72°C. DNA was precipitated overnight and resuspended in 50 µL of milliQ water. A 5 µL DNA aliquot of the LM-PCR was analyzed on a 1.2% agarose gel to check the size (average size of 300–400 bp) and PCR efficiency. The PCR product was used as a template for the radioactive labelling reaction. The reaction mixture contained 5–15 µL of amplified DNA in 50 µL (1x DNA pol buffer, 2 mM $MgCl_2$, 0.2 mM dATP, dTTP and dGTP, 25 µM dCTP, 1 µM oJW102, 0.8 µCi/µL α-33P-dCTP and 5 U DNA polymerase). The mix was denatured for 5 minutes at 95°C, annealed 5 minutes at 50°C, and amplified for 30 minutes at 72°C. The reaction product was purified through a ProbeQuant G-50 column (Amersham Biosciences, Piscataway, NJ, USA) to remove unincorporated α-33P-dCTP and oligonucleotides.

## Total mRNA extraction

Total RNA isolated from budding yeast cells was prepared as described [22], but using a multiple-sample automated device (Fast-Prep®) to break cells.

## Hybridisation, image analysis and data normalization

Nylon filters were made using PCR-amplified whole ORF sequences as probes. After pre-hybridizing nylon membranes for 1h in 5X SSC, 5X Denhart's, 0.5% SDS, 100 µg/mL salmon sperm DNA, hybridizations were performed using 3–5 mL of the same solution containing labelled RNA. Hybridizations were conducted during 40–44 h in a roller oven at 65°C. After hybridization, filters were washed once in 2X SSC, 0.1% SDS for 30 min, and twice in 0.2X SSC, 0.1% SDS for 30 min. Filters were exposed for up to 7 days to an imaging plate (BAS-MP, FujiFilm) which was read at 50 µm resolution in a phosphorimager scanner (FLA-3000, FujiFilm). Both GRO and RNAp ChIP experiments were performed in triplicate

by swapping the filters in each replicate among the different sampling times, as described in [20] Images were quantified by using the ArrayVision 7.0 software (Imaging Research, Inc.). The signal intensity for each spot was the background subtracted ARM Density (Artifact Removed Median). Only those values that were 1.35 times above the corresponding background were taken as valid measurements. To compare the RNAp ChIP data between experiments, the median binding ratio of the 32 rDNA spots were arbitrarily set as background. Reproducibility of the replicates was checked using the ArrayStat software (Imaging Research, Inc.). Normalization between conditions was done using the global median method for RNAp ChIP and for GRO experiments. The ratio between immunoprecipitated (IP) and whole cell extract (WCE) in each experiment (or No-Ab and WCE) after normalization was taken as the binding ratio.

### Data analyses

The changes in RNAp ChIP and GRO data for all genes were evaluated using cluster analysis of log2-transformed and z-score normalized values. The cluster composition was analyzed with WebMeV (Multiple Experiment Viewer) [35], employing the SOTA clustering procedure [23]. Expression data from [17] was averaged across two cycles and z-score normalized. Decay rates from the same source were averaged across samples, log-transformed, and scaled to z-scores. The relationship between GRO and RNAp ChIP was modeled using a linear model without an intercept term. Values for gene categories or cell cycle phases were summarized by the mean, with error bars representing the 95% confidence interval as estimated by 1,000 bootstrap resamples.

## Results

### Elongating RNA polymerase II is highly dynamic during the cell cycle

To determine if the activity of elongating RNAP II undergoes changes throughout the cell cycle, we used alpha-factor to synchronize *S. cerevisiae* populations at START, the decision point within G1 in which cells commit to a new round of cell division [24]. We then released cells from the arrest and took samples for total RNAP II (ChIP) or active RNAP II (Genomic run-on) at 0, 30 and 60 minutes (see scheme in Fig 1A). FACS profile and budding index were used to estimate the average cell cycle stage of each sample (S1 Fig). Almost all cells from samples at time 0 were in G1, whereas most cells in the 60 min samples were in G2/M. Cells from the 30 min samples were in between, with at least one third of them exhibiting the characteristic small bud of S phase, indicating a clear enrichment in this phase.

First analyses revealed many genes in which the levels of active and total RNAP II change during the cell cycle (S2A Fig). As shown in Fig 1B, global correlation between total RNAP II (ChIP) and active RNAP II (GRO) was significant and oscillated across the cell cycle, showing a maximum Pearson correlation coefficient ($R^2$) of 0.41 at the 30 min timepoint after START. When we focused in the subset of genes with significant periodic regulation across the cell cycle [18], correlation between total and active RNAP II was slightly higher in G1 ($R^2$ value of 0.38 versus 0.31 in all genes) and in S-G2 ($R^2$ value of 0.44 versus 0.41) (Fig 1B).

We analyzed RNAP II behavior in the different categories of cyclic genes, according to its phase of maximal expression (S1 Table). We found rather parallel RNAP II ChIP and GRO average profiles (Fig 1C). Using the previously published data from Cramer's lab [17], we also found that these transcriptional patterns were similar to their corresponding mRNA profiles, and symmetrical to those of mRNA decay (Fig 1C, see later).

This parallelism between total and active RNAP II patterns in cyclic genes supports the technical consistency of our experimental data. However, some subtle differences between the RNAP II ChIP and GRO average profiles were detected, particularly in the G1-S and M genes (Fig 1C). Moreover, the modest global correlation of RNAP II ChIP and GRO across the genome (Fig 1B) seemed to indicate that the control of elongation dynamics might contribute to cell cycle regulation. We hypothesized that changes in gene expression during the cell cycle may result from a variety of strategies in which transcription would be regulated, not only by increasing or decreasing the number of RNAP II molecules engaged on a gene, but also by changing their catalytic activity during elongation.

## Some cycling genes are transcriptionally regulated at the elongation step

To uncover these putative regulatory strategies, we clustered all genes according to their RNAP II ChIP and GRO profiles along the cell cycle. We followed the SOTA procedure [23], using the linear correlation coefficient as the distance metric between genes and a variability threshold of 40% as training conditions (S2B Fig). This resulted in ten distinct clusters, named A to J, which divided the gene set into consistent groups (Fig 2A, S1 Table). Average GRO and RNAP II ChIP signals for each cluster are shown in Fig 2B, alongside mRNA expression and decay data from [17] (see later).

Some clusters exhibited parallel profiles for total (ChIP) and active RNAP II (GRO). For instance, cluster C showed significant decreases in both RNAP II ChIP and GRO signals at the 30-minute time point, followed by a minimal change at 60 minutes (Fig 2B). A similarly simple regulation was observed in cluster G, which displayed a sustained increase in both RNAP II ChIP and GRO signals after G1 (Fig 2B). Finally, cluster J exhibited peaks of both RNAP II ChIP and GRO at the 30-minute time point (Fig 2B). In all three of these clusters, regulation is likely exerted by controlling RNAP II recruitment, with little or no modulation of its elongation activity.

In contrast, the other seven clusters exhibited different patterns of average RNAP II ChIP and GRO signals, with some of them showing completely dissimilar profiles (Fig 2B). For example, cluster B displayed a marked decrease in RNAP II ChIP at 60 minutes after G1, while showing a peak in GRO at 30 minutes (Fig 2B). Cluster H showed increased RNAP II ChIP signals at both 30 and 60 minutes, but a flat pattern in GRO (Fig 2B). The opposite was observed in cluster F, where RNAP II ChIP signals increased slightly at 30 minutes, whereas GRO signals dropped at this time point and remained low at 60 minutes (Fig 2B). Cluster A also exhibited a distinct pattern: a flat RNAP II ChIP profile with a strong increase in GRO after G1, peaking at the 30-minute time point (Fig 2B). Unparallel profiles were also observed in clusters D and I, where RNAP II ChIP signals decreased and increased, respectively, 30 minutes after G1, without comparable changes in their GRO profiles (Fig 2B).

We next examined whether these alternative patterns of RNAP II elongation were significant for the expression of cycling genes in the ten clusters, as defined by Cyclebase [18]. We found significant enrichment, either positive or negative, in most clusters. Positive enrichments were particularly clear in clusters D and E (G1 genes), G (M genes), I (S/G2 and G2/M genes), and J (S, S/G2, and G2/M genes) (Fig 2C). Additionally, when we examined the presence of our clusters within previously defined cycling gene subgroups, we found significant enrichment of clusters D, E, and F in G1 genes, and of cluster G in M genes (S3 Fig). Interestingly, we also observed significant exclusion of all clusters except B, G, I, and J from G2/M genes (S3 Fig). Furthermore, clusters D, E, and F were significantly excluded from S/G2 genes (S3 Fig).

Notably, some clusters displaying dissimilar RNAP II ChIP and GRO profiles across the cell cycle, such as D, F and I, were associated with cycling genes in the analyses mentioned above. GRO and RNAp ChIP profiles of several G1-expressed genes belonging to cluster D are shown in S4A Fig. Examples of G1- and G1/S-expressed genes from cluster F, and S/G2- and G2/M genes from cluster I are also shown (S4B–S4C Figs). These results suggest that regulation of RNAP II activity during transcription elongation is compatible with cell cycle control of gene expression, establishing a link between transcription elongation dynamics and cell cycle regulation.

We also investigated whether the genes within each cluster shared specific transcriptional activators or repressors, based on previous experimental evidence. We used the "Rank by TF" tool from the Yeastrack database [25]. We found that most clusters were significantly enriched in genes regulated by specific activators or repressors (Fig 2D). For instance, cluster G was enriched in genes regulated by Spt10, Spt2, and Hif1, all of which are known to control histone gene transcription in S phase [26], as well as by Thi2, an activator of thiamine pyrophosphate biosynthesis genes, which are upregulated during mitosis [16] (Fig 2D).

## Regulation of RNA polymerase II elongation explains the cycling pattern of ribosome biogenesis genes

We wondered whether the genes in each cluster, in addition to sharing common transcriptional regulation, also exhibited functional similarities. Gene ontology (GO) analysis confirmed that this was indeed the case (S2 Table).

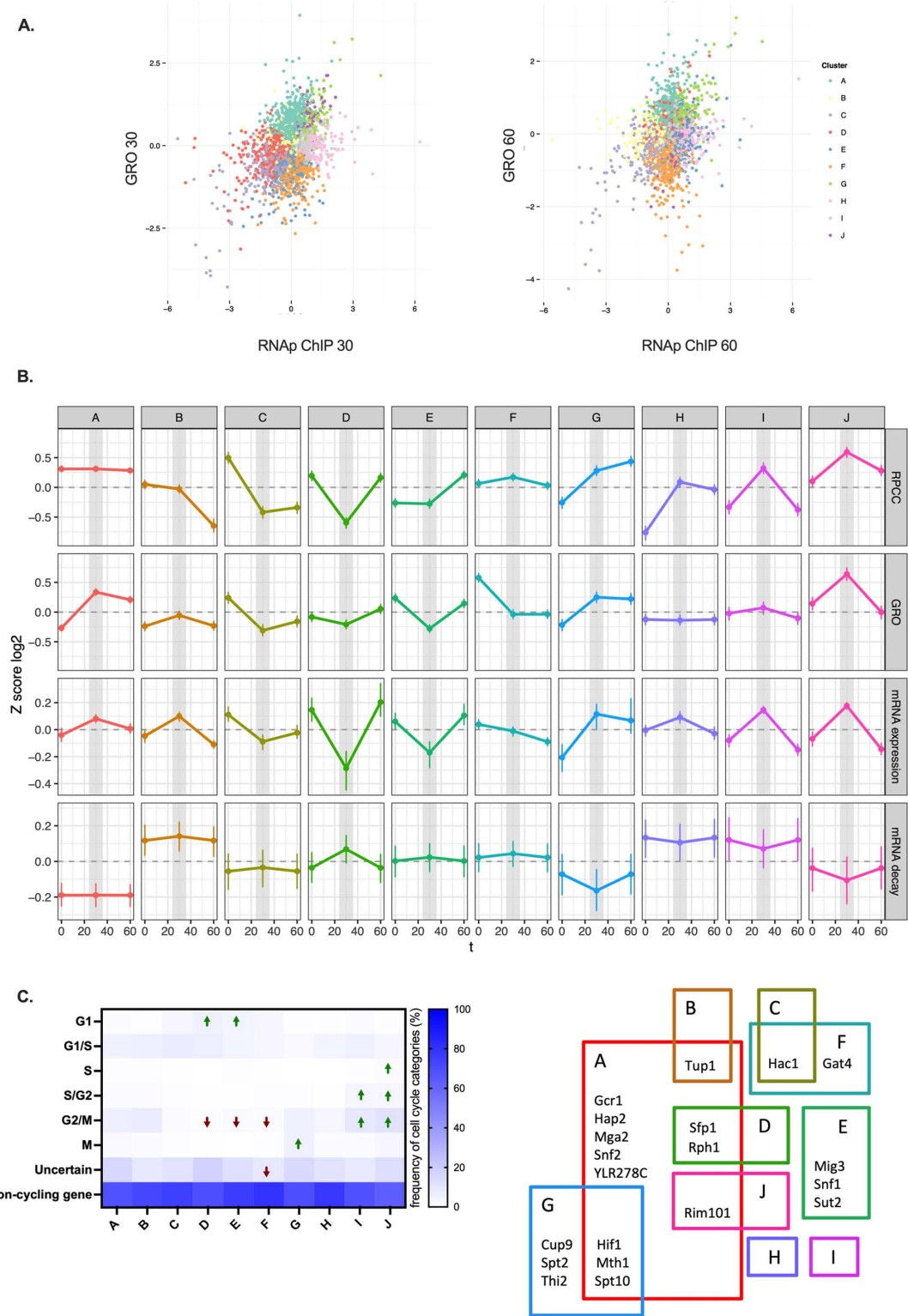

**Fig 2. Clustering analysis of total and active RNAP II at genes during the transition from G1 to G2.** A) Overview of the different clusters identified through SOTA analysis. B) Behavior of total RNAP II (RNAp ChIP) and active RNAP II (GRO) in each identified cluster. mRNA expression and mRNA decay data from Eser et al. [17] are also presented. Genes included in each cluster, along with all transcriptomic data, are listed in S1 Table. To

determine the proportion of active polymerases, we performed a linear regression analysis using GRO as the dependent variable and RNAp ChIP as the predictor, without an intercept term. The resulting regression coefficient represents the fraction of RNA polymerases actively engaged in transcription. By interpreting this coefficient as the efficiency of polymerase engagement, we estimate the percentage of active polymerases relative to the total chromatin-bound population. C) Distribution of cell cycle genes according to their peak of maximal expression (CellCycleBase 3.0) [18] in each of the clusters identified. Green arrows indicate groups in which the enrichment or depletion of cell cycle genes is statistically significant (logistic regression with Benjamini-Hochberg correction). D) Transcription factors known to regulate genes belonging to the clusters described in A. Only transcription factors significantly enriched in one or two clusters (Yeastrack [25], hypergeometric test $p < 10^{-5}$), according to DNA binding or expression evidence, are shown. Other transcription factors significantly enriched in three or more clusters include: Bas1, Cst6, Tec1, Gat1, Cbf1, Met28, Pdr1, Gcn4, Met31, Met32, Pdr3, Rpn4, and Yap1.

For example, cluster A was enriched in terms related to ribosome biogenesis and the structural constituents of ribosomes (ribosomal proteins) (Fig 3A). These categories correspond to the so-called "RiBi" and "RP" regulons, respectively [17]. Consistent with these findings, we observed that cluster A genes were enriched in the global activator Gcr1, known to act as a coactivator of Rap1 in the regulation of ribosomal protein genes [27] (Fig 2D). A significant proportion of cluster A genes also appeared to be targets of Sfp1 and Rph1 (Fig 2D). These two regulators mediate the control of ribosomal protein genes in response to nutrient and stress signaling [28,29], and their molecular functions are connected to transcription elongation. Sfp1 promotes RNA pol II backtracking in its target genes [30,31], while Rph1, a histone H3-K36 demethylase, is linked to chromatin dynamics in the context of RNAP II elongation [32].

Cluster D was enriched in a wide set of categories related to respiration and metabolic pathways localized to mitochondria (Fig 3B). Interestingly, genes in cluster D were also enriched in specific targets of Sfp1 and Rph1 (Fig 2D).

Given the high significance of their GO categories, we focused on cluster A and the ribosome biogenesis genes, whose regulation is critical for cell growth (reviewed in [33]; see also [34]). We analyzed the cluster distribution of the 207 genes from the RiBi regulon present in our dataset and found that more than 50% belonged to cluster A, confirming the significance of this enrichment. In contrast, most other clusters were either not enriched or significantly depleted of this regulon (Fig 3C). All RiBi regulon subgroups, including genes involved in nucleotide metabolism, translation factors, and tRNA synthetases, were highly represented in cluster A (Fig 3C).

We also examined the average GRO and RNAp ChIP profiles of the entire RiBi regulon. As in cluster A, between the 0 and 30-minute time points, the GRO signal increased more than the RNAp ChIP signal (Fig 3D). For the RP regulon, less dissimilar average profiles of total and active RNAP II were observed (Fig 3D). This difference from the cluster A average can be explained by the fact that ribosomal protein genes were not only enriched in cluster A but also in cluster G (Fig 3C), which shows a similar GRO profile to cluster A but with a parallel RNAp ChIP profile (Fig 2B). Ribosomal protein genes in cluster G mainly encode components of the large ribosomal subunit (Fig 3C). The significance of this finding is unclear, but it may reflect the existence of different regulatory mechanisms for the synthesis of each ribosomal subunit.

Moreover, the differences between the RP and RiBi regulons are consistent with the overall correlation between GRO and RNAp ChIP signals for these gene groups across the cell cycle. While ribosomal protein genes exhibited very high $R^2$ values at all three time points (between 0.89 and 0.94), RiBi genes showed a drastic increase in correlation from 0 ($R^2 = 0.40$) to 30 minutes ($R^2 = 0.70$) (Fig 3E).

Ribosome-related mRNAs have not been systematically described as undergoing cyclic regulation [18], although a recent study supports the periodic regulation of genes related to rRNA synthesis, cytoplasmic translation, and ribosome biogenesis at the proteomic level [14]. We analyzed mRNA expression of a subset of eight genes from the RiBi and RP regulons by RT-PCR in alpha factor-synchronized cells. All analyzed RiBi genes exhibited peaks of expression 20 minutes after release from alpha factor (S5 Fig). This cycling pattern was not as clear in the three ribosomal protein genes analyzed (S5 Fig). We conclude that most genes of the RiBi regulon control their expression across the cell cycle by altering RNAP II activity during transcription elongation.

**A.**

| Term ID | Term Name | $p_{adj}$ (query_1) ↑ |
|---|---|---|
| GO:0042254 | ribosome biogenesis | $3.879\times10^{-34}$ |
| GO:0032991 | protein-containing complex | $4.802\times10^{-29}$ |
| GO:0005488 | binding | $4.464\times10^{-20}$ |
| GO:0043412 | macromolecule modification | $1.560\times10^{-10}$ |
| GO:0003735 | structural constituent of ribosome | $6.916\times10^{-6}$ |
| GO:0140098 | catalytic activity, acting on RNA | $1.510\times10^{-4}$ |
| GO:0071042 | nuclear polyadenylation-dependent mRNA cat... | $6.064\times10^{-3}$ |
| GO:0071051 | polyadenylation-dependent snoRNA 3'-end pr... | $6.064\times10^{-3}$ |
| GO:0009156 | ribonucleoside monophosphate biosynthetic pr... | $6.605\times10^{-3}$ |
| GO:0008170 | N-methyltransferase activity | $8.650\times10^{-3}$ |
| GO:0004298 | threonine-type endopeptidase activity | $9.054\times10^{-3}$ |
| GO:0043021 | ribonucleoprotein complex binding | $1.107\times10^{-2}$ |
| GO:0051028 | mRNA transport | $2.655\times10^{-2}$ |
| GO:0006450 | regulation of translational fidelity | $2.829\times10^{-2}$ |
| GO:0051664 | nuclear pore localization | $3.334\times10^{-2}$ |
| GO:0010494 | cytoplasmic stress granule | $4.081\times10^{-2}$ |

**B.**

| Term ID | Term Name | $p_{adj}$ (query_1) |
|---|---|---|
| GO:0016491 | oxidoreductase activity | $3.892\times10^{-3}$ |
| GO:0016743 | carboxyl- or carbamoyltransferase activity | $1.291\times10^{-2}$ |
| GO:0009055 | electron transfer activity | $4.596\times10^{-2}$ |
| GO:0008652 | amino acid biosynthetic process | $3.904\times10^{-4}$ |
| GO:0009084 | glutamine family amino acid biosynthetic proce... | $1.663\times10^{-3}$ |
| GO:0009987 | cellular process | $1.006\times10^{-2}$ |
| GO:0042773 | ATP synthesis coupled electron transport | $1.210\times10^{-2}$ |
| GO:0098798 | mitochondrial protein-containing complex | $1.416\times10^{-4}$ |
| GO:0005759 | mitochondrial matrix | $2.736\times10^{-3}$ |
| GO:0005622 | intracellular anatomical structure | $5.570\times10^{-3}$ |
| GO:0005750 | mitochondrial respiratory chain complex III | $2.516\times10^{-2}$ |
| GO:0031090 | organelle membrane | $2.919\times10^{-2}$ |
| GO:0070469 | respirasome | $4.943\times10^{-2}$ |

**C.**

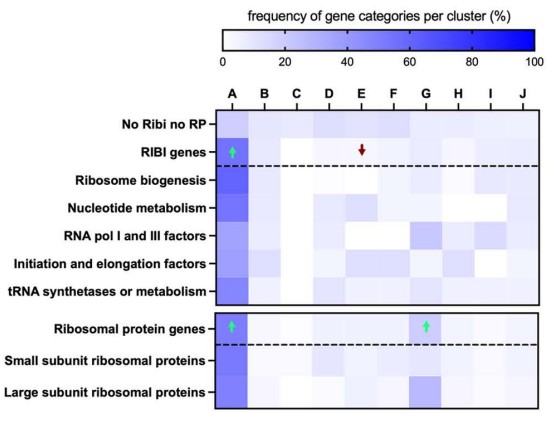

**D.**

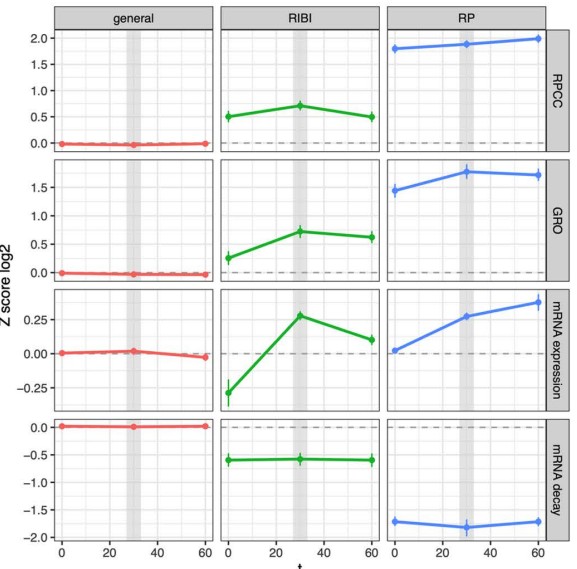

**E.**

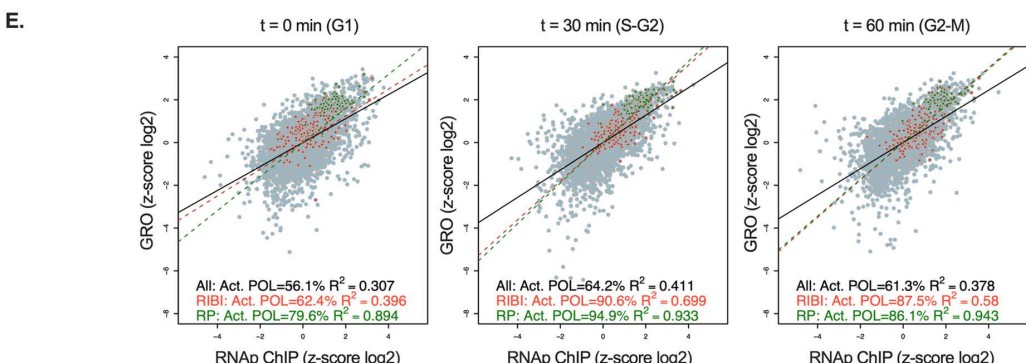

**Fig 3. Cell cycle and functional analysis of total and active RNAP II across the cell cycle.** A) Gene ontology analysis with gProfiler for genes in cluster A. B) Similar analysis for genes in cluster D. C) Distribution of identified clusters and logistic regression analyses within the RIBI regulon and subcategories (upper section) and within the RP regulon and subcategories (lower section). D) Representation of total RNAP II (RNAp ChIP) and active

RNAP II (GRO) in RiBi and RP genes. mRNA expression and mRNA decay data from Eser et al. [17] are also represented. E) Normalized values for all genes (grey), RiBi regulon (red), and RP regulon (green) for total RNAP II (RNAp ChIP) and active RNA Pol II (GRO). The proportion of active polymerases was calculated as described in Fig 1B.

### RNA polymerase II elongation patterns define alternative gene expression strategies during the cell cycle

The RiBi regulon is a good example where GRO profiles anticipated mRNA expression patterns better than RNAP II ChIP (Fig 3D). To extend this perspective, we represented the average mRNA levels of these clusters, using the accurate transcriptomic quantification of synchronized cells after START performed by Eser et al. [17]. We confirmed that, in most cases, mRNA expression profiles followed GRO values more closely than RNAP II ChIP (compare the RNAP II ChIP, GRO, and mRNA expression panels in Fig 2B).

However, two clear exceptions were detected. Clusters D and I, which are enriched in G1-expressed genes and S/G2/M genes, respectively (Fig 2C), showed a closer parallelism of mRNA expression profiles to RNAP II ChIP than to GRO (Fig 2B). These two cases were puzzling, since RNAP II activity measured by GRO should be a better proxy for transcription than total RNAP II present on genes, as the latter includes both backtracked RNAP II molecules and actively elongating ones [8,31,35,36].

To understand these cases, we incorporated mRNA decay values along the cell cycle, which were obtained by Cramer's laboratory [17] using a metabolic labeling approach. As previously described, this additional information aids in understanding the regulatory behavior of cycling genes. For example, the decrease in mRNA expression of G1/S genes at the 30-minute time point is better explained by increased mRNA decay than by the relatively flat profile of transcriptional activity measured by GRO (Fig 1C).

This additional layer of information allowed us to explain the divergence between GRO and mRNA expression profiles detected in clusters D and I. We found that average mRNA decay values in cluster D were maximal at 30 minutes (Fig 2B). Therefore, the combination of a mild decrease in transcriptional activity (GRO) with a marked increase in mRNA decay explains the strong downregulation of mRNA expression levels for this group of genes at the 30-minute time point (Fig 2B).

In contrast, cluster I showed a mild increase in transcriptional activity (GRO) combined with a slight decrease in mRNA decay, which explains the comparatively higher increase in mRNA expression at 30 minutes (Fig 2B). In fact, comparing clusters I and J, which showed nearly identical mRNA expression profiles, is illustrative of two alternative strategies that result in the same average mRNA expression pattern. In cluster I, mRNA expression regulation is achieved by comparable changes in RNAP II activity and mRNA decay, whereas in cluster J, regulation of RNAP II activity was much more pronounced (Fig 2B). This difference is likely due to the overall higher stability (lower decay) of mRNAs in cluster J compared to cluster I (Fig 2B). As expected, transcriptional regulation of stable mRNAs involves more intense changes in transcriptional activity than for unstable mRNAs. GRO changes also triggered stronger alterations of mRNA expression in ribosome biogenesis genes than in ribosomal protein genes, which exhibited much more stable mRNAs (Fig 3D).

The comparison of clusters D and E is also very informative. These two sets of genes shared similar overall mRNA stabilities, intermediate between those of clusters I and J (Fig 2B). In this context, the stronger regulation of transcriptional activity shown by cluster E was insufficient to match the regulatory effect in mRNA levels observed in cluster D, which combined milder control of transcription activity with a more pronounced change in mRNA stability (Fig 2B).

A similar conclusion was reached when comparing clusters A (stable mRNAs) and H (very unstable mRNAs). In this case, drastic upregulation of transcriptional activity with no regulation of mRNA decay (cluster A) produced the same average mRNA expression pattern as no change in transcriptional activity combined with slight downregulation of decay (cluster H).

The synergistic effect produced by the combined control of transcription and decay was also evident when we compared clusters A and G, both of which express stable mRNAs. These two clusters exhibited upregulated GRO profiles after START; however, in the case of cluster G, its combination with mRNA decay downregulation led to stronger upregulation of mRNA levels than in cluster A, where no significant change in decay was detected (Fig 2B). Thus, in all ranges of mRNA stability, the most pronounced regulation of mRNA levels was achieved by concerted control of RNAP II activity and mRNA decay.

## Discussion

The comparison of total RNAP II occupancy on gene bodies, as measured by ChIP (RNAP II ChIP), with transcriptional activity assessed by genomic run-on (GRO) allowed us to test whether the regulation of transcription elongation plays a significant role in gene expression across the cell cycle. Only 21% of the 4,764 analyzed genes were classified into clusters that exhibited parallel RNAp ChIP and GRO profiles after alpha factor release (Fig 2B and S2 Fig), indicating that most budding yeast genes undergo some form of regulation at the RNAP II activity level.

Several gene clusters were significantly enriched in genes previously shown to exhibit a cyclic pattern of mRNA expression. This was the case for cluster G, which is associated with M-specific genes and their transcription regulators (Fig 2C–2D), and which displayed parallel RNAp ChIP, GRO, and mRNA expression profiles (Fig 2B). No alterations in transcriptional activity during elongation are required to explain the regulatory behavior of these genes.

In contrast, clusters D (enriched in G1-specific genes) and I (enriched in S-G2-M genes) displayed dissimilar RNAp ChIP and GRO profiles (Fig 2B, S4 Fig), indicating that some alterations in RNAP II activity must occur during transcription elongation, likely through RNAP II backtracking. In these two cases, GRO profiles alone did not explain the mRNA expression patterns, unless mRNA decay was also considered (Fig 2B, see later).

Interestingly, cluster A, which also exhibited non-parallel RNAp ChIP and GRO average profiles, contained more than 50% of the RiBi regulon (Fig 3A). Although these genes were not previously described as periodic [18], we confirmed this regulation for some examples through RT-PCR (S5 Fig) and found a regulated mRNA expression pattern for the entire ribosome biogenesis group in the cell-cycle expression dataset produced by Eser et al. [17] (Fig 3D).

RNAP II activity in ribosomal protein and ribosome biogenesis genes has previously been shown to be regulated during elongation in response to environmental stimuli mediated by protein kinase A [8]. Interestingly, another regulon capable of modulating its elongating RNAP II activity in response to these conditions is composed of genes related to respiration and other mitochondrial functions [8]. GO analyses revealed significant enrichment of this type of genes in cluster D (Fig 3B).

We have previously described that these regulons change their expression in response to cell growth in opposite directions: mRNA concentrations of ribosomal protein and ribosome biogenesis genes increase with the growth rate, whereas mRNA concentrations of mitochondria-related genes decrease, in line with the low respiratory activity of fast-growing yeast cultures [37]. These controls occur through different mechanisms: ribosomal protein and ribosome biogenesis genes upregulate transcription with cell growth without overall changes in mRNA stability, while mitochondria-related genes decrease mRNA stability with increasing growth rate, without significant alterations in transcription rates [37,38]. In agreement with this, the regulation of mRNA levels in cluster A, and more specifically in ribosome biogenesis and ribosomal protein genes, appears to be primarily transcriptional (Figs 2B and 3E), whereas the mRNA expression in cluster D across the cell cycle requires mRNA decay to be considered (Fig 2B).

The decreased expression of mitochondria-related genes at the 30-minute time point, when at least one third of the cells were in S phase, aligns with published evidence suggesting that DNA replication occurs during the reductive phase of the *S. cerevisiae* metabolic cycle [39,40]. The increased expression of ribosome biogenesis genes at this time point is likely related to the S-phase, as nucleolus biogenesis is coupled with rDNA replication [41]. Moreover, the synthesis of a complete set of new RNA polymerase I and ribosome components, particularly biogenesis factors, is required before the complete nucleolar reorganization that occurs in mitosis [42].

As discussed earlier, D and I are the two clusters where: i) alterations in RNAP II activity, likely due to backtracking, are occurring, and ii) mRNA stability is playing a significant role in regulation. The fact that genes clustered according to elongating RNAP II activity features (RNAp ChIP/GRO profiles) show distinct mRNA decay profiles underscores the importance of transcriptional elongation dynamics in gene expression across the cell cycle. This was especially evident when comparing clusters I and J, which display identical average mRNA expression profiles but differ in mRNA stabilities (Figs 2B).

A molecular connection between RNAP II elongation dynamics and the stability of transcribed mRNA was recently described by us for genes controlled by the transcription factor Sfp1 [30]. During transcription of these genes, Sfp1 promotes RNAP II backtracking until late elongation, when it is transferred from the polymerase to the nascent mRNA, which then becomes imprinted for higher stability [30]. It is tempting to hypothesize that a similar mechanism could be operating in the genes of clusters D and I. In fact, genes in cluster D are significantly enriched in targets of Sfp1 (Fig 2D). In this regard, the mRNA stability of mitochondria-related genes (enriched in cluster D, Fig 2B) is controlled by Puf3, a factor that has been proposed to bind its targets co-transcriptionally [43].

In addition to Puf3-regulated ones, other genes whose mRNAs are targeted by RNA binding proteins that control their stability also exhibit transcriptional activation/repression, indicating the existence of specific mechanisms of crosstalk, which sum up effects on synthesis and decay rates as a way to enhance regulation [44].

These mechanisms of crosstalk may involve complex interactions between transcriptional regulators and RNA-binding proteins that are dynamically modulated across the cell cycle. Such interactions not only influence gene expression but also fine-tune the stability of the mRNA transcripts, contributing to the precise regulation required for cellular processes such as growth and division. Indeed, the balance between transcriptional activity and mRNA decay is crucial for maintaining the appropriate levels of gene products at different stages of the cell cycle.

## Supporting information

**S1 Fig. Cell cycle distribution in assessed samples.** Representative example of the budding index and the FACS profile of samples in which stages in which active and total RNA pol II were analysed.
(PDF)

**S2 Fig. Distribution of total RNA polymerase II (RNAp ChIP) and active RNA polymerase II (GRO) at genes during the transition from G1 to G2.** A) Plot representing total RNA pol II and active RNA pol II present in each gene at each time point analysed. B) Plot representing clusters identified with SOTA.
(PDF)

**S3 Fig. Distribution of identified clusters within the different categories of cell cycle genes according to their peak of maximal accumulation (CellCycle Base 3.0).** Green arrows represent groups in which the enrichment or depletion of clusters is statistically significant (logistic regression with BF correction).
(PDF)

**S4 Fig. Some periodically expressed genes exhibit dissimilar GRO and RNAp ChIP profiles along the cell cycle.** A) Examples of G1-expressed genes (*AMS1, ARA1, CHS1, GPD1, TPS1* and *YDL023C*) belonging to cluster D. B) Examples of G1- (*ALD4*, *HXT4* and *HXT5*) and G1/S-expressed genes (*IDH2*, *HEM15* and *YLR327C*) from cluster F. C) Examples of S/G2- (*ARP10*) and G2/M-expressed genes (*FKH2, STB5, TUB2, YOR246C* and *YTH1)* from cluster I.
(PDF)

**S5 Fig. Cell cycle regulation of RiBi and RP genes.** Gene ontology analysis highlights a strong enrichment of cluster A in RP and RIBI genes. Manual validation of mRNA levels obtained for selected RiBi and RP genes in synchronized cells released from a G1 arrest with alpha factor (10 minutes interval sampling).
(PDF)

**S1 Table. Experimental data.** GRO, RNAp ChIP, mRNA expression and mRNA decay data for all genes, indicating the cluster where each genes is located, whether the gene has been previously described as cell-cycle regulated, and whether it belongs to RiBi or RP regulons. mRNA expression and decay data are from Cramer´s lab [17]. (XLSX)

**S2 Table. Gene ontology analyses.** Results of the gene ontology analyses performed with each gene cluster. (XLSX)

## Acknowledgments

We thank all members of the IBiS Gene Expression and Ribosome Biogenesis Labs for their helpful scientific discussions, and all technicians from the IBiS and CITIUS facilities for their experimental assistance.

## Author contributions

**Conceptualization:** Jesús de la Cruz, José Enrique Pérez-Ortín, María de la Cruz Muñoz-Centeno, Sebastián Chávez.

**Data curation:** Douglas Maya-Miles, José García-Martínez, Ildefonso Cases.

**Formal analysis:** Douglas Maya-Miles, José García-Martínez, Ildefonso Cases.

**Funding acquisition:** María de la Cruz Muñoz-Centeno.

**Investigation:** Douglas Maya-Miles, José García-Martínez, Rocío Pasión.

**Methodology:** Jesús de la Cruz, José Enrique Pérez-Ortín, María de la Cruz Muñoz-Centeno.

**Supervision:** María de la Cruz Muñoz-Centeno.

**Writing – original draft:** Sebastián Chávez.

**Writing – review & editing:** Douglas Maya-Miles, Jesús de la Cruz, José Enrique Pérez-Ortín, María de la Cruz Muñoz-Centeno, Sebastián Chávez.

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
