## [Decision Letter · Decision Letter 0]

11 Feb 2025

PONE-D-25-00223Regulation of transcription elongation anticipates alternative gene expression strategies across the cell cyclePLOS ONE

Dear Dr. Chávez,

Thank you for submitting your manuscript to PLOS ONE. After careful consideration, we feel that it has merit but does not fully meet PLOS ONE’s publication criteria as it currently stands. Therefore, we invite you to submit a revised version of the manuscript that addresses the points raised during the review process.

After considering the review from one referee and my own review of the paper, the main concern is the presentation of the study. This manuscript requires additional text to adequately guide the reader through the study and results. Please address all the comments raised by the reviewer. In addition, please avoid the use of abbreviations as much as possible for clarity. For example, use of "RNAP II ChIP" rather than RPCC and "ribosome biogenesis" in place of RiBi. For the purpose of review, 1.5 line spacing is desirable as are page numbers. 

We look forward to receiving your revised manuscript.

Kind regards,

Barbara Jennings

Academic Editor

PLOS ONE

Journal Requirements:

2. Thank you for stating the following financial disclosure: [This work was supported by grants PID2020-112853GB-C31 to JEP- O, PID2022-136564NB-I00 to JdlC and PID2020-112853GB-C32 to MCM-C funded by MCIN/AEI/10.13039/501100011033/ERDF, EU.]

3. Thank you for stating the following in the Acknowledgments Section of your manuscript: [We thank all members of the IBiS Gene expression and Ribosome biogenesis labs for helpful scientific discussion, and all technicians from IBiS and CITIUS facilities for their experimental help. This work was supported by grants PID2020-112853GB-C31 to JEP- O, PID2022-136564NB-I00 to JdlC and PID2020-112853GB-C32 to MCM-C funded by MCIN/AEI/10.13039/501100011033/ERDF, EU.]

Please remove any funding-related text from the manuscript and let us know how you would like to update your Funding Statement. Currently, your Funding Statement reads as follows: [This work was supported by grants PID2020-112853GB-C31 to JEP- O, PID2022-136564NB-I00 to JdlC and PID2020-112853GB-C32 to MCM-C funded by MCIN/AEI/10.13039/501100011033/ERDF, EU.]

4. In the online submission form, you indicated that your data is available only on request from a third party. Please note that your Data Availability Statement is currently missing the contact details for the third party, such as an email address or a link to where data requests can be made]. Please update your statement with the missing information.

5. Please include captions for your Supporting Information files at the end of your manuscript, and update any in-text citations to match accordingly. Please see our Supporting Information guidelines for more information: http://journals.plos.org/plosone/s/supporting-information .

Reviewers' comments:

Reviewer's Responses to Questions

**Comments to the Author**

1. Is the manuscript technically sound, and do the data support the conclusions?

Reviewer #1: Yes

2. Has the statistical analysis been performed appropriately and rigorously? 

Reviewer #1: Yes

3. Have the authors made all data underlying the findings in their manuscript fully available?

Reviewer #1: No

4. Is the manuscript presented in an intelligible fashion and written in standard English?

Reviewer #1: No

5. Review Comments to the Author

Reviewer #1: A majority of studies dealing with understanding transcriptional regulation are focused on the initiation step. The transcription, however, is also regulated at the elongation and termination steps, and this downstream regulation is rather a neglected area of research. The focus of this investigation in elucidating regulation of transcription at the elongation step, therefore, is appropriate. Authors employ RNAPII ChIP and GRO approaches to study transcription at different stages of cell cycle in budding yeast. Their results suggest that genes coding for ribosomal components and linked to ribosome biogenesis are expressed at a higher level in pre-mitotic phases and are subjected to regulation at the elongation step. They also uncovered the role of mRNA stability in the overall expression of a number of these genes. These findings are significant and make an interesting scientific story. It is, however, difficult for a general reader of PLOS One to grasp and appreciate these results the way it is presented in this manuscript. Authors need to improve the presentation of results to make it more easily understandable. The following suggestions may help:

Major Points

1. Authors mention that some of the cycling genes are subjected to regulation at the elongation step. Inclusion of a list of these genes, and their RNAPII ChIP as well as GRO profile may help strengthen conclusions of the study.

2. Figure 1C is confusing. The X-axis shows time point after release from alpha-factor arrest, while Y-axis depicts Z-scores. It is, however, not clear what the headings on the top of every panel, G1, G1/S, S, S/G2, and M means. Does this mean the profile shown below is for genes expressed in the indicated stage of cell cycle? This needs to be clarified in the legend and also in the text. If G1 means the profile below is for genes predominantly expressed in G1 phase, then identity of these genes must be included in the form of a table in the supplementary data section.

3. In Figures 2B and S2B, authors need to explain in detail the basis of grouping genes in nine clusters. Just describing that clustering was done on the basis of RPCC and GRO profiles is not sufficient. Describe the range of RPCC and GRO signal values used for grouping genes in each cluster. Briefly explain SOTA while describing Figure 2A.

4. In Figure 2C, authors show distribution of gene clusters expressed in different stages of cell cycle. It will be beneficial for readers if these genes can be listed in the form of a table in supplementary data section.

Minor Points

1. In the third paragraph of ‘Introduction’, it needs to be clarified that TFIIS is not the only factor that regulates elongation. There are other factors, especially chromatin remodeling factors, histone chaperone and histone modifying enzymes that also regulate elongation through chromatin.

2. What is START? Explain for the benefit of readers not familiar with yeast.

3. Explain significance of R2 value.

4. In ‘Discussion’ section, briefly explain why mitochondrial related genes exhibit decreased mRNA stability with growth rate.

6. PLOS authors have the option to publish the peer review history of their article (what does this mean? ). If published, this will include your full peer review and any attached files.

**Do you want your identity to be public for this peer review?** For information about this choice, including consent withdrawal, please see our Privacy Policy .

Reviewer #1: No

---

## [Author Response · Author response to Decision Letter 1]

21 Mar 2025

PONE-D-25-00223

Regulation of transcription elongation anticipates alternative gene expression strategies across the cell cycle

PLOS ONEDear Dr. Chávez,Thank you for submitting your manuscript to PLOS ONE. After careful consideration, we feel that it has merit but does not fully meet PLOS ONE’s publication criteria as it currently stands. Therefore, we invite you to submit a revised version of the manuscript that addresses the points raised during the review process.After considering the review from one referee and my own review of the paper, the main concern is the presentation of the study. This manuscript requires additional text to adequately guide the reader through the study and results. Please address all the comments raised by the reviewer. In addition, please avoid the use of abbreviations as much as possible for clarity. For example, use of "RNAP II ChIP" rather than RPCC and "ribosome biogenesis" in place of RiBi.

Response: RPCC has been replaced by RNAP II ChIP across the manuscript and abbreviated as RNAp ChIP in the figures. The use of RiBi was been limited to the name of the “RiBi regulon”, which is composed not only by ribosome biogenesis factors but also by other co-regulated genes.

For the purpose of review, 1.5 line spacing is desirable as are page numbers. 

Response: Done.

If applicable, we recommend that you deposit your laboratory protocols in protocols.io to enhance the reproducibility of your results. Protocols.io assigns your protocol its own identifier (DOI) so that it can be cited independently in the future. For instructions see: https://journals.plos.org/plosone/s/submission-guidelines#loc-laboratory-protocols. Additionally, PLOS ONE offers an option for publishing peer-reviewed Lab Protocol articles, which describe protocols hosted on protocols.io. Read more information on sharing protocols at https://plos.org/protocols?utm_medium=editorial-email&utm_source=authorletters&utm_campaign=protocols.We look forward to receiving your revised manuscript.

Kind regards,

Barbara JenningsAcademic EditorPLOS ONE

Reviewers' comments:Reviewer's Responses to Questions

Comments to the Author1. Is the manuscript technically sound, and do the data support the conclusions?The manuscript must describe a technically sound piece of scientific research with data that supports the conclusions. Experiments must have been conducted rigorously, with appropriate controls, replication, and sample sizes. The conclusions must be drawn appropriately based on the data presented.

Reviewer #1: Yes

2. Has the statistical analysis been performed appropriately and rigorously?

Reviewer #1: Yes

3. Have the authors made all data underlying the findings in their manuscript fully available?The PLOS Data policy requires authors to make all data underlying the findings described in their manuscript fully available without restriction, with rare exception (please refer to the Data Availability Statement in the manuscript PDF file). The data should be provided as part of the manuscript or its supporting information, or deposited to a public repository. For example, in addition to summary statistics, the data points behind means, medians and variance measures should be available. If there are restrictions on publicly sharing data—e.g. participant privacy or use of data from a third party—those must be specified.

Reviewer #1: No

Response: All experimental data utilized to produce the figures of this manuscript are now supplied in the supplementary tables.

4. Is the manuscript presented in an intelligible fashion and written in standard English?PLOS ONE does not copyedit accepted manuscripts, so the language in submitted articles must be clear, correct, and unambiguous. Any typographical or grammatical errors should be corrected at revision, so please note any specific errors here.

Reviewer #1: No

Response: English has been thoroughly corrected.

5. Review Comments to the AuthorPlease use the space provided to explain your answers to the questions above. You may also include additional comments for the author, including concerns about dual publication, research ethics, or publication ethics. (Please upload your review as an attachment if it exceeds 20,000 characters).

Reviewer #1: A majority of studies dealing with understanding transcriptional regulation are focused on the initiation step. The transcription, however, is also regulated at the elongation and termination steps, and this downstream regulation is rather a neglected area of research. The focus of this investigation in elucidating regulation of transcription at the elongation step, therefore, is appropriate. Authors employ RNAPII ChIP and GRO approaches to study transcription at different stages of cell cycle in budding yeast. Their results suggest that genes coding for ribosomal components and linked to ribosome biogenesis are expressed at a higher level in pre-mitotic phases and are subjected to regulation at the elongation step. They also uncovered the role of mRNA stability in the overall expression of a number of these genes. These findings are significant and make an interesting scientific story. It is, however, difficult for a general reader of PLOS One to grasp and appreciate these results the way it is presented in this manuscript. Authors need to improve the presentation of results to make it more easily understandable. The following suggestions may help:Major Points1. Authors mention that some of the cycling genes are subjected to regulation at the elongation step. Inclusion of a list of these genes, and their RNAPII ChIP as well as GRO profile may help strengthen conclusions of the study.

Response: Following the recommendation of the reviewer, we now show in the new Supplementary figure 4, some examples of periodically expressed genes with dissimilar GRO and RNAp ChIP profiles from clusters D, F and I. The new Supplementary table 1 also contains RNAPII ChIP and GRO data for all genes, including previously defined cycling genes, which are identified according to their expression cell-cycle stage.

2. Figure 1C is confusing. The X-axis shows time point after release from alpha-factor arrest, while Y-axis depicts Z-scores. It is, however, not clear what the headings on the top of every panel, G1, G1/S, S, S/G2, and M means. Does this mean the profile shown below is for genes expressed in the indicated stage of cell cycle?

Response: Exactly. This is what it means.

This needs to be clarified in the legend and also in the text. If G1 means the profile below is for genes predominantly expressed in G1 phase, then identity of these genes must be included in the form of a table in the supplementary data section.

Response: We have changed the labelling of the figure to facilitate its understanding. We have also improved the description of these results in the text. The new Supplementary table 1 lists the genes included in each category.

3. In Figures 2B and S2B, authors need to explain in detail the basis of grouping genes in nine clusters. Just describing that clustering was done on the basis of RPCC and GRO profiles is not sufficient. Describe the range of RPCC and GRO signal values used for grouping genes in each cluster. Briefly explain SOTA while describing Figure 2A.

Response: The number of clusters is arbitrary. However, we decided to set it to ten because it produced the most consistent distribution of patterns. Lower number of clusters would have resulted in too high internal diversity; higher number would have produced duplicated clusters with minor differences. SOTA is now better described in the text (page 13).

4. In Figure 2C, authors show distribution of gene clusters expressed in different stages of cell cycle. It will be beneficial for readers if these genes can be listed in the form of a table in supplementary data section.

Response: The genes included in each cluster are listed in the new Supplementary table 1.

Minor Points1. In the third paragraph of ‘Introduction’, it needs to be clarified that TFIIS is not the only factor that regulates elongation. There are other factors, especially chromatin remodeling factors, histone chaperone and histone modifying enzymes that also regulate elongation through chromatin.

Response: As requested by the reviewer, we have clarified this point in the introduction, including references that describe the whole set of transcription elongation factors.

2. What is START? Explain for the benefit of readers not familiar with yeast.

Response: START is the decision point within G1 in which cells commit to a new round of cell division. We have explained this at the beginning of the Results section and cited the landmark paper by Lee Hartwell where it was defined.

3. Explain significance of R2 value.

Response: R2 is the square of the Pearson correlation coefficient, also known as the coefficient of determination. It represents the proportion of the variance in the dependent variable that can be explained by the independent variable in a linear regression model. We now explained the meaning the first time that it is mentioned in the text (second paragraph of the Results section).

4. In ‘Discussion’ section, briefly explain why mitochondrial related genes exhibit decreased mRNA stability with growth rate.

Response: Due to the low respiratory activity of fast-growing yeast cultures, mitochondrial-related genes become downregulated. We have previously shown [37] that this downregulation takes place by decreasing mRNA stability. This is now more clearly explained in the Discussion section.

---

## [Decision Letter · Decision Letter 1]

1 Apr 2025

Regulation of transcription elongation anticipates alternative gene expression strategies across the cell cycle

PONE-D-25-00223R1

Dear Dr. Chávez,

We’re pleased to inform you that your manuscript has been judged scientifically suitable for publication and will be formally accepted for publication once it meets all outstanding technical requirements.

Kind regards,

Barbara Jennings

Academic Editor

PLOS ONE

Additional Editor Comments (optional):

Reviewers' comments:

Reviewer's Responses to Questions

**Comments to the Author**

1. If the authors have adequately addressed your comments raised in a previous round of review and you feel that this manuscript is now acceptable for publication, you may indicate that here to bypass the “Comments to the Author” section, enter your conflict of interest statement in the “Confidential to Editor” section, and submit your "Accept" recommendation.

Reviewer #1: All comments have been addressed

2. Is the manuscript technically sound, and do the data support the conclusions?

Reviewer #1: Yes

3. Has the statistical analysis been performed appropriately and rigorously? 

Reviewer #1: Yes

4. Have the authors made all data underlying the findings in their manuscript fully available?

Reviewer #1: Yes

5. Is the manuscript presented in an intelligible fashion and written in standard English?

Reviewer #1: Yes

6. Review Comments to the Author

Reviewer #1: (No Response)

7. PLOS authors have the option to publish the peer review history of their article (what does this mean? ). If published, this will include your full peer review and any attached files.

**Do you want your identity to be public for this peer review?** For information about this choice, including consent withdrawal, please see our Privacy Policy .

Reviewer #1: No

---

## [Editor Report · Acceptance letter]

PONE-D-25-00223R1

PLOS ONE

Dear Dr. Chávez,

I'm pleased to inform you that your manuscript has been deemed suitable for publication in PLOS ONE. Congratulations! Your manuscript is now being handed over to our production team.

Kind regards,

on behalf of

Dr. Barbara Jennings

Academic Editor

PLOS ONE